# S-1 versus Doublet Regimens as Adjuvant Chemotherapy in Patients with Advanced Gastric Cancer after Radical Surgery with D2 Dissection—A Propensity Score Matching Analysis

**DOI:** 10.3390/cancers12092384

**Published:** 2020-08-23

**Authors:** Meng-Che Hsieh, Shih-Ho Wang, Ching-Ting Wei, Chung-Yen Chen, Yen-Yang Chen, Sung-Nan Pei, Yu-Fen Tsai, Kun-Ming Rau

**Affiliations:** 1Department of Hematology-Oncology, E-Da Cancer Hospital, Kaohsiung 822445, Taiwan; ed111215@edah.org.tw (M.-C.H.); ed112124@edah.org.tw (S.-N.P.); ed112116@edah.org.tw (Y.-F.T.); 2College of Medicine, I-Shou University, Kaohsiung 822445, Taiwan; ed106224@edah.org.tw (C.-T.W.); ed104874@edah.org.tw (C.-Y.C.); 3Division of General Surgery, Department of Surgery, Kaohsiung Chang Gung Memorial Hospital and Chang Gung University College of Medicine, Kaohsiung 822445, Taiwan; cphtem44@gmail.com; 4Division of General Surgery, Department of Surgery, E-Da Hospital, Kaohsiung 822445, Taiwan; 5Division of Hematology-Oncology, Department of Internal Medicine, Kaohsiung Chang Gung Memorial Hospital and Chang Gung University College of Medicine, Kaohsiung 822445, Taiwan; cyy@cgmh.org.tw

**Keywords:** adjuvant chemotherapy, gastric cancer, prognosis, survival, stage, lymph node ratio

## Abstract

Background: Fluoropyrimidine- and platinum-based doublet regimen is the standard treatment of adjuvant chemotherapy (AC) for gastric cancer (GC). Our study aims to compare S1 with doublet regimens as AC in patients with advanced GC after radical surgery with D2 dissection. Methods: Patients who were diagnosed with GC and underwent a curative surgery with D2 dissection followed by AC were enrolled into our study. A propensity score matching analysis was performed to reduce the selection bias. Kaplan–Meier curves were estimated for recurrence-free survival (RFS) and overall survival (OS). Cox regression models were conducted for survival. Results: After propensity sore matching, 64 patients with S1 and 64 patients with doublet regimens were identified. The median RFS (*p* = 0.355) and OS (*p* = 0.309) were both insignificant between S1 and ST. Cox regression analysis demonstrated that pathologic stage and lymph node ratio (LNR) were independently correlated with survival. Patients were then stratified into low risk and high risk groups. The median RFS (*p* < 0.001) and OS (*p* < 0.001) had significant differences between low risk and high risk. In the high-risk group, doublet regimens were strongly associated with survival (*p* = 0.020) as compared with S1. While in the low-risk group, doublet regimen and S1 did not have statistically different survival benefits. Conclusions: Our study demonstrated that doublet regimens are superior to S1 in high-risk groups, and that survival outcomes are similar between doublet regimens and S1 in low-risk groups. Our prognostic model might have clinical implications for AC.

## 1. Introduction

Gastric cancer (GC) is the leading malignancy in the upper digestive tract. It is the fourth most common human malignant disease, and the second most frequent cause of cancer-related death worldwide, accounting for 6.8% of all cancer diagnoses and 8.8% of all cancer-related deaths [1]. In 2012, there were 951,000 new GC patients and 723,000 deaths due to GC [2]. To date, the only curative treatment is radical surgery with D2 dissection. However, the results are still unsatisfactory, owing to the high recurrent and metastatic rate [3]. The overall 5-year survival rate in the western world remains low with apparently 80% for stage I, less than 50% for stage II and less than 20% for stage III GC patients [4]. To improve the poor outcome, many studies have examined various aspects of surgical techniques, including extended lymph node dissection, the addition of radiotherapy as well as the effect of adjuvant chemotherapy (AC) or chemoradiotherapy [5]. Several established evidences show that surgery followed by AC for GC could improve survival over surgery alone [6,7,8]. The pivotal study (CLASSIC) investigated the effect on the disease-free survival of AC with capecitabine plus oxaliplatin after D2 gastrectomy, compared with D2 gastrectomy only in patients with stage II–IIIB gastric cancer, and demonstrated that the 5-year overall survival (OS) rate was 78% (95% confidence interval (CI) 0.74–0.82) in the adjuvant capecitabine and oxaliplatin (XELOX) group versus 69% (95% CI 0.64–0.73) in the observation group [9,10]. AC with 5-fluorouracil and cisplatin also showed a statistically significant survival benefit when compared with surgery alone. The Cox model with hazard ratio (HR) was 0.74 [95% CI 0.54–1.02; *p* = 0.063] for death and 0.70 (95% CI 0.51–0.97; *p* = 0.032) for recurrence [11]. Based on these results, fluoropyrimidine- and platinum- based doublet regimens are standard AC in patients with GC.

S1 is an oral form of fluoropyrimidine. A phase III study (ACTS-GC) evaluated S1 as an AC for Japanese patients who had undergone curative D2 gastrectomy for stage II or III GC, and identified that the 3-year OS rate was 80.1% in the S1 group and 70.1% in the surgery-only group (HR: 0.68, 95% CI, 0.52–0.87; *p* = 0.003) [12,13]. Therefore, AC with S1 is now also another treatment for GC patients in Japan. However, an ACTS-GC trial only demonstrated that GC patients had better survival benefits with S1 than those with observation. Little was known regarding the efficacy of S1 in comparison with standard fluoropyrimidine- and platinum-based doublet regimens. Our study aims to compare S1 with doublet regimens as an AC in patients with advanced GC after radical surgery with D2 dissection.

## 2. Materials and Methods

### 2.1. Patients’ Selection

Patients who were diagnosed with GC from 2010 to 2017 at Kaohsiung Chang Gung Memorial Hospital and Kaohsiung E-Da Hospital were reviewed. Inclusion criteria were age older than 18 years, pathologically confirmed gastric adenocarcinoma, a curative surgery with D2 dissection for GC as a primary treatment followed by AC. Chemotherapy regimen was limited to S1 or fluoropyrimidine- and platinum-based doublet regimens. AC regimens were decided at the physicians’ discretion. Computed tomography was arranged periodically every 3–4 months for evaluation about the disease. After a retrospective chart review, a total of 181 patients were enrolled into our study for an outcomes evaluation. This study was approved by the Chang Gung Memorial Hospital Institutional Review Board (No. 201600980B0), and this work was exempt from requiring consent.

### 2.2. Chemotherapy Protocol

The doublet group included Capecitabine plus oxaliplatin (XELOX) and 5-fluorouracil plus cisplatin (FP). In the S1 group, patients were treated with 80 mg/m² S1 orally twice per day for 14 days followed by 7 days rest for 12 months. In the XELOX group [14], patients were treated with a 2-week cycle of 85 mg/m² intravenous oxaliplatin on day 1 of each cycle plus 1000 mg/m² oral capecitabine for 10 days of each cycle for 6 months. In the FP group, patients were treated with a 4-week cycle of 100 mg/m² cisplatin and 4-days of continuous infusion of 1000 mg/m² 5- fluorouracil per day on day 1–4 of each cycle for 6 months. The prophylactic medication consisted of an antiemetic and hydration.

### 2.3. Statistical Analysis

All the clinical variables were collected from a medical chart review and calculated with frequencies. Chi-square tests were used to assess the differences between groups for categorical variables. Lymph node ratio (LNR) was defined as the ratio of the number of positive lymph nodes to the number of total lymph nodes removed. The median LNR in our patients was 0.21. A propensity score matching analysis was performed to reduce the selection bias. The oncologic outcomes were presented with recurrence-free survival (RFS) and overall survival (OS). RFS was calculated as the time from surgery to recurrence or death. OS was calculated as the time between surgery and death or the last visiting. Kaplan–Meier curves were estimated for RFS and OS. We conducted a log-rank test with Cox regression models using “enter” selection to adjust for the effects of potential confounders. All *P* values were two sided and considered to be statistically significant if *p* values < 0.05.

## 3. Results

### Patients’ Characteristics

The median age was 55 years and median follow-up interval was 41 months. During the follow-up period, 35% of our patients died, and cancer was the main reason of their death. Table 1 summarized the clinical characteristics of our patients. Statistically, most patients were younger than 65 years (64%), and were male (66%). The pathologic differentiation was, predominately, poorly differentiated/signet ring carcinoma (PD/SRC) (72%), rather than well or moderately differentiated (W/MD) (28%). Approximately 77% of our patients had stage III disease and 23% of our patients had stage II disease with lymphovascular invasion, which led them to receive AC. After propensity score matching, 64 patients with S1 and 64 patients with doublet chemotherapy were recruited for survival estimation. The baseline characteristics, including age, gender, performance status, pathologic differentiation, pathologic stage and LNR, were all matched between these two groups. As for doublet regimens, 42 patients received XELOX and 22 patients received FP as an AC.

The survival curves of RFS and OS are plotted in Figure 1. The median RFS was 56.8 months in the S1 group and not reach (NR) in the doublet group (*p* = 0.355). The 5-year survival rates were 70% versus 78% in S1 and doublet group, respectively (*p* = 0.309). Cox regression analyses with survival for potential prognostic factors were performed. The hazard ratio (HR) with 95% CIs is depicted in Table 2. Multivariate analysis demonstrated that stage (HR: 0.03, 95% CI: 0.01–0.27, *p* = 0.003 for RFS, HR: 0.03, 95% CI: 0.01–0.63, *p* = 0.024 for OS) and LNR (HR: 0.47, 95% CI: 0.27–0.82, *p* = 0.008 for RFS, HR: 0.37, 95% CI: 0.19–0.75, *p* = 0.005 for OS) were independent predictors that correlated with RFS and OS, respectively. The adjuvant chemotherapy regimens, S1 or doublet, did not have a significant impact on survival.

According to our multivariate analyses, we divided our patients into a low-risk group and a high-risk group by counting the number of significantly prognostic variables, including stage and LNR. Patients with zero to one prognostic factor were classified as low-risk, while those with both factors were classified as high-risk. The low-risk group contained 70 patients and the high-risk group contained 58 patients. Subgroup analyses of RFS and OS based on risk factors are presented in Figure 2. The median RFS was 52.4 months and NR in the high-risk group and low-risk group, respectively (*p* < 0.001). The 5-year survival rates were 81% and 59% in the low-risk group and high-risk group, respectively (*p* < 0.001). Furthermore, our study focused on the impact of chemotherapy regimens on the survival of the low-risk and high-risk group. Figure 3 shows the survival curves of the low-risk group and high-risk group, stratified by S1 and doublet regimens. The median OS were insignificant between S1 and doublet regimens in low risk groups, while the median OS was significant longer with doublet regimens than those with S1 in the high-risk group. In the low-risk group, the 5-year survival rates were 84% and 82% for doublet regimens and S1, respectively (*p* = 0.698), while in high risk group, the 5-year survival rates were 72% and 42% for doublet regimens and S1, respectively (*p* = 0.02).

## 4. Discussion

This is a real world retrospective comparison between S1 and doublet regimens as an AC for advanced gastric cancer after D2 radical surgery. For decades, AC has been proven to provide survival benefits for patients with advanced gastric cancer after radical surgery, as compared with surgery alone [15]. Therefore, current guidelines recommend that fluoropyrimidine- and platinum-based doublet regimens are the standard adjuvant regimen for GC [16]. However, there were no randomized control studies comparing the oncologic outcomes of different AC regimens. After propensity score matching, our study suggests that the OS was insignificantly different between S1 and doublet regimens. Moreover, we constructed a prognostic model by stage and LNR, and separated patients into low-risk and high-risk groups. In consideration of chemotherapy regimens, the median OS was insignificant between S1 and doublet regimens in low risk groups, while the median OS was significantly longer with doublet regimens than those with S1 in high-risk groups. Previous literature is consistent with our results. Subgroup analysis of the CLASSIC study demonstrated greater treatment benefits with doublet chemotherapy in patients with a higher stage [9,10], while subgroup analysis of an ACTS-GC study suggested a smaller treatment benefit with S1 monotherapy in high-risk GC patients [12,13]. Our conclusion may be useful in clinical counseling and survival prediction. Further prospective studies with larger sample sizes are warranted to confirm our results.

Recently, Yoshida et al. reported the promising results of the JACCRO GC-07 study [17]. JACCRO GC-07 is a randomized phase III study which compared postoperative S1 plus docetaxel (DS) over S1 alone for R0 resection of pathologic stage III gastric cancer. The second interim analysis showed the superiority of DS (66%) to S1 (50%) for a 3-year RFS (HR, 0.632; 99.99% CI, 0.400 to 0.998; *p* < 0.001). Another encouraging interim analysis of the ARTIST-2 study was also published by Park et al. in 2019 [18]. The ARTIST-2 study analyzed the prognosis of adjuvant S1 monotherapy for one year, S1 plus oxaliplatin (SOX) for six months, or SOX plus chemoradiotherapy (SOXRT) in patients with pathologically-staged II or III, node-positive, D2-resected GC patients. RFS in S1 was significantly shorter than in SOX and SOXRT arms (HR: S-1 vs. SOX, 0.617, *p* = 0.016 and S-1 vs. SOXRT, 0.686, *p* = 0.057). The RFS at 3-years was found to be 65%, 78% and 73% in S-1, SOX and SOXRT arms, respectively. These results are similar to our finding that the oncologic outcomes of doublet regimen are better than those of S1 monotherapy in selected GC patients, such as more advanced-stage and high-risk groups. However, whether DS or SOX are superior to fluoropyrimidine- plus platinum-based doublet regimens remains unresolved. Nakamura et al. evaluated two phase 2 trials to indirectly compare SOX with XELOX and suggested comparable efficacy of SOX and XELOX in stage 3 GC patients after D2 gastrectomy [19]. Further prospective randomized study is warranted to answer this question.

Our study enrolled more GC patients with PD/SRC, up to 72%, which is much higher than those in the CLASSIC study and other proceeding literature. Histopathologic type is recognized as a prognostic factor in GC in several preceding studies [20]. It is well known that GC with PD/SRC usually has worse outcomes than other pathologic types, especially in advanced stages and metastatic disease [21,22]. Most studies regarding this issue come from Asia, where epidemiology and disease biology are different from Western countries [21]. The treatment guidelines of the Japanese Gastric Cancer Association also recommend that the histopathologic type is among the indicators for treatment of GC [23]. Emerging data also demonstrates that patients with PD/SRC are generally younger in age, more often female in gender, higher in grade and higher in TNM stage than those with other histological types [24,25]. Kim et al. conducted a retrospective analysis and found lymph node metastasis to be the only independent prognostic factor in GC with PD/SRC [25]. Increasing evidence observed that GC represents multiple separate diseases with the same generic pathology but distinct pathogenesis [26]. PD/SRC are considered biologically distinct from other pathologic types, which themselves differ from tumor location and degree of differentiation [27]. Therefore, it is clinically meaningful to develop a more effective treatment for this kind of differentiation.

Several poor prognostic factors were also identified for advanced GC after D2 radical surgery followed by AC. In the pivotal CLASSIC study, subgroup analysis demonstrated that age and nodal status was significantly correlated with RFS and OS [10]. In et al. analyzed 1687 GC patients after suboptimal lymphadenectomy receiving AC. They concluded that higher Charlson score, ≥15 lymph nodes examined, higher tumor grade, and tumor location in the gastric cardiac were factors associated with significantly decreased OS [8]. Aoyama collected 103 GC patients treated with AC, and verified that body weight loss of 15% was regarded as a significant factor for survival [28]. Ema et al. retrospectively reviewed 152 GC patients treated with S1 as an AC, and demonstrated that a lymph node ratio ≥16.7% and intestinal-type histology were significant as predictors of prognosis, independent from the pathological stages [29]. Meanwhile, stage-stratified PD/SRC was also proven to have a prognostic role in advanced GC patients [21,25]. Identical to the abovementioned publication, our study suggests that stage and LNR are independent prognostications for survival. By counting these two factors, we created a useful predictive model for advanced GC patients receiving AC.

Our study is a real word retrospective analysis with several potential limitations. Therefore, propensity score matching was used to diminish the bias. The chemotherapy regimen was decided at the discretion of a physician, which might be a major source of bias in this study. All the patients enrolled in this study were Asian. Because the oncologic outcomes of AC in Asian patients may differ from those in western countries, the applicability of our results to patients treated in the west is unknown. Meanwhile, one single institutional experience, the small size of our cohort, and the inconsistent follow-up duration also limit the power of our study. A more recent retrospective study regarding AC in stage III gastric cancer was also consistent with our conclusion that there was no statistical difference in OS between S1 and doublet regimens [30]. However, they did not compare the clinical characteristics of the S1 group with those of doublet regimens. Our study demonstrated the direct comparison of oncologic outcomes between S1 and doublet regimens. We also constructed a prognostic model by counting stage and LNR. Based on our results, our work may have clinical implications for advanced GC patients receiving AC. Given the inevitable selection biases, which are inherent to any retrospective study, more prospective studies with larger cohorts are warranted to validate our results.

## 5. Conclusions

This is a real world retrospective study to compare the oncologic outcomes between S1 and doublet regimens as an AC in advanced GC patients after radical surgery with D2 dissection. Based on our results, we disclosed that S1 seems to have a comparable efficacy to doublet regimens. We also constructed a prognostic model including stage and LNR for advanced GC patients. In high-risk groups, doublet regimens are superior to S1, while in low-risk groups, the survival outcomes are similar between doublet regimens and S1. These results may have clinical implications for physicians who treat GC patients receiving AC. Further validation with a prospective study with a larger sample size is warranted.

## Figures and Tables

**Figure 1 cancers-12-02384-f001:**
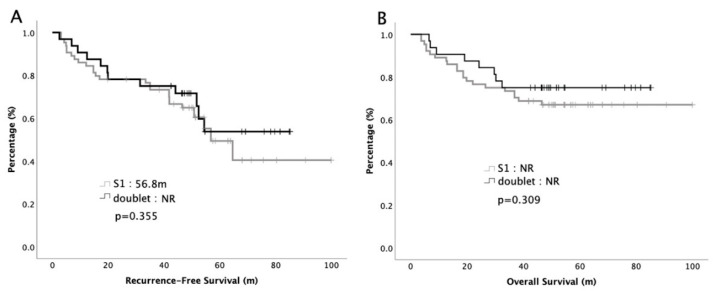
Recurrent-free survival (**A**) and overall survival (**B**) of 128 patients with advanced gastric cancer receiving adjuvant chemotherapy.

**Figure 2 cancers-12-02384-f002:**
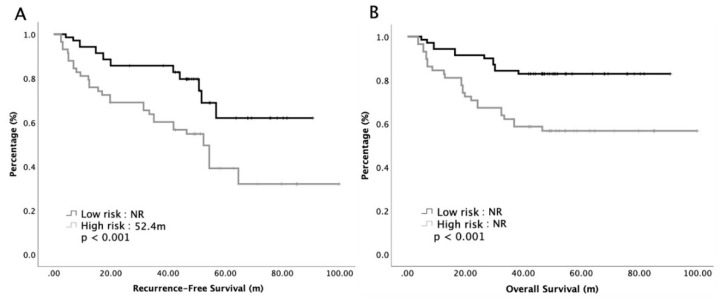
Recurrent-free survival (**A**) and overall survival (**B**) of 128 patients with advanced gastric cancer receiving adjuvant chemotherapy, stratified by low risk or high risk.

**Figure 3 cancers-12-02384-f003:**
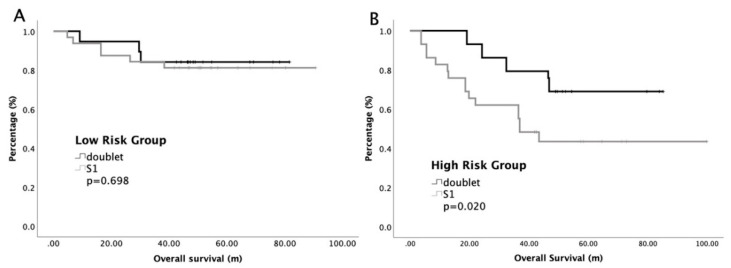
Overall survival of advanced gastric cancer patients with low risk (**A**) or high risk (**B**), stratified by adjuvant chemotherapy regimens.

**Table 1 cancers-12-02384-t001:** Baseline patient characteristics of 181 patients with advanced gastric cancer receiving adjuvant chemotherapy, stratified by PSM.

Variables	Before PSM			After PSM		
	Doublet	S1	*p* Value	Doublet	S1	*p* Value
	N = 117	N = 64		N = 64	N = 64	
Age			0.023			0.476
≤65	82 (70%)	34 (53%)		38 (59%)	34 (53%)	
>65	35 (30%)	30 (47%)		26 (41%)	30 (47%)	
Gender			0.034			0.277
Male	84 (72%)	36 (56%)		42 (66%)	36 (56%)	
Female	33 (28%)	28 (44%)		12 (34%)	28 (44%)	
Performance status			0.008			0.57
0–1	97 (83%)	42 (66%)		45 (70%)	42 (66%)	
2–3	20 (17%)	22 (34%)		19 (30%)	22 (34%)	
Differentiation			0.133			1
W/MD	28 (24%)	22 (34%)		22 (34%)	22 (34%)	
PD/SRC	89 (76%)	42 (66%)		42 (66%)	42 (66%)	
Surgery			0.758			0.639
Total gastrectomy	20 (17%)	12 (19%)		10 (16%)	12 (19%)	
Subtotal gastrectomy	97 (83%)	52 (81%)		54 (84%)	52 (81%)	
Pathologic T stage			0.214			0.639
1–2	10 (9%)	12 (19%)		10 (16%)	12 (19%)	
3–4	107 (91%)	52 (81%)		54 (84%)	52 (81%)	
Pathologic N stage			0.371			0.211
0–1	28 (24%)	18 (28%)		12 (19%)	18 (28%)	
2–3	89 (76%)	46 (72%)		52 (81%)	46 (72%)	
Pathologic stage			0.676			0.676
II	26 (22%)	16 (25%)		14 (22%)	16 (25%)	
III	91 (78%)	48 (75%)		50 (78%)	48 (75%)	
LN ratio			0.042			0.111
≤0.21	51 (43%)	38 (59%)		29 (45%)	38 (59%)	
>0.21	66 (57%)	26 (41%)		35 (55%)	26 (41%)	
median LN removed	15	18		16	18	

PSM–propensity score match; W/MD–well/moderately differentiated; PD/SRC–poorly differentiated/signet ring cell; LN, lymph node.

**Table 2 cancers-12-02384-t002:** Cox regression analysis of parameters associated with survival.

Variables	RFS		OS	
	HR (95% CI)	*p* Value	HR (95% CI)	*p* Value
Age, ≤65 vs. >65	0.84 (0.49–1.43)	0.517	0.86 (0.45–1.67)	0.663
Gender, Male vs. Female	0.55 (0.31–1.19)	0.145	0.68 (0.34–1.35)	0.265
PS, 0–1 vs. 2–3	0.63 (0.36–1.08)	0.090	0.64 (0.33–1.33)	0.183
Differentiation, W/MD vs. PD/SRC	0.55 (0.30–1.03)	0.062	0.75 (0.37–1.52)	0.424
Stage II vs. III	0.03 (0.01–0.27)	0.003	0.03 (0.01–0.63)	0.024
LNR, ≤0.21 vs. >0.21	0.47 (0.27–0.82)	0.008	0.37 (0.19–0.75)	0.005
Regimen, S1 vs. doublet	0.78 (0.45–1.33)	0.360	0.72 (0.37–1.37)	0.313

RFS–recurrence-free survival, OS–overall survival, W/MD–well/moderate differentiation, PD/SRC–poorly differentiation/signet ring cell, LNR–lymph node ratio, HR–hazard ratio, CI–confidence interval.

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
