# Peer review of "S-1 versus Doublet Regimens as Adjuvant Chemotherapy in Patients with Advanced Gastric Cancer after Radical Surgery with D2 Dissection—A Propensity Score Matching Analysis"

_cancers, 2020, doi:10.3390/cancers12092384_

Round 1
Reviewer 1 Report
I thank the authors having made the suggested amendments,
They have provided the reference for biweekly xelox in their reply. I would suggest they insert this citation in the methods section when they describe their xelox regimen as this is not commonly used so it would be helpful for readers to understand the basis of their treatment regimen.
Author Response
Thank you for your suggest. I had inserted the citation in "method" section ( as in Page 2. Line 93.)
Reviewer 2 Report
The paper has been improved in this revised form
Author Response
Thank you for your kind review.
This manuscript is a resubmission of an earlier submission. The following is a list of the peer review reports and author responses from that submission.
Round 1
Reviewer 1 Report
Properly designed methodology and analysis of results.
Introduction: references for adjuvant chemo and chemoradiation can be improved. eg. GASTRIC metanalysis JAMA 2010, INT0116. More contemporaneous references should include JACCRO GC07 (JCO 2019) and the interim analysis of the ARTIST-2 study (JCO 2019).
Methods:
1) exclusion criteria of incomplete pre-planned courses of AC can introduce bias due by excluding patients on regimens which are more toxic and discontinue treatment early.
2) the doublet control arm XELOX (is a biweekly regimen, which is different from the standard XELOX regimen as described in the CLASSIC study). reference is needed to support its equivalence. The 5-FU cisplatin doublet is generally not a recommended adjuvant regimen as this has not been shown to be superior to observation alone following surgery. Hence, comparison of these 2 doublet regimens with the standard TS-1 monotherapy is challenging.
Results:
1) i would suggest providing data on the % of total and subtotal gastrectomy. The median number of LN removed during surgery in both arms.
2) Using LNR is interesting and the authors have provided references where this has been used. However, as there is no generally accepted LNR cut-off. I would suggest the authors provide a breakdown of Stage 3A, 3B and 3C rather than a single category of > Stage II. The use of Node positive vs negative GC or Stage II vs Stage III would allow a more generalisable comparison.
Discussion
1) this should include the JACCRO GC-07 and ARTISTS 2 study (presented interim results) which have showed superiority of doublet chemo for recurrence free survival compared to TS-1 monotherapy.
2) It is worth discussing the subgroup analysis of CLASSIC study which suggested greater treatment benefit with XELOX with higher N stage, as well as the subgroup analysis of ACTS-GC study which suggested smaller treatment benefit in Node positive GC. These comparisons are relevant to the conclusions of your study.
Reviewer 2 Report
Authors evaluated the S1 versus doublet regimens as adjuvant chemotherapy in patients with advanced gastric cancer after radical surgery with D2 dissection in 128 patients. median RFS and OS were comparable in the two groups. Stage and lymph node ratio were independent predictors of survival and stratified two populations into low (0-1) and high (2) risk group. In high risk group survival was significantly better in patients treated with doublet regimens.
The study is interesting, but I have some major concerns.
-This is a retrospective study and adjuvant regimens were decided by physicians' discretions. This point should be further explained because a risk of selection bias is high. Moreover, doublet regimens included capecitabine plus oxaliplatin and 5-fluorouracil plus cisplatin: the distribution of patients in the different regimens should be added in Table 1.
- Authors concluded that S1 had a comparable efficacy with doublet regimens especially in low risk group. In my opinion, the main result of the study is the demonstration of two risk groups of patients with different survival after adjuvant therapy and this point should be stressed in Abstract and Conclusion.
Minor point
-Table and figure numbers are not reported in the text.